# Impact of Outpatient Rehabilitation Service in Preventing the Deterioration of the Care-Needs Level Among Japanese Older Adults Availing Long-Term Care Insurance: A Propensity Score Matched Retrospective Study

**DOI:** 10.3390/ijerph16071292

**Published:** 2019-04-10

**Authors:** Michio Maruta, Takayuki Tabira, Hyuma Makizako, Akira Sagari, Hironori Miyata, Koji Yoshimitsu, Gwanghee Han, Kazuhiro Yoshiura, Masahiro Kawagoe

**Affiliations:** 1Medical Corporation, Sansyukai, Okatsu Hospital, 3-9, Masagohommachi, Kagoshima 890-0067, Japan; 2Department of Occupational Therapy, School of Health Sciences, Faculty of Medicine, Kagoshima University, 8-35-1, Sakuragaoka, Kagoshima 890-8544, Japan; tabitaka@health.nop.kagoshima-u.ac.jp (T.T.); yoshimitsu@health.nop.kagoshima-u.ac.jp (K.Y.); 3Department of Physical Therapy, School of Health Sciences, Faculty of Medicine, Kagoshima University, 8-35-1, Sakuragaoka, Kagoshima 890-8544, Japan; makizako@health.nop.kagoshima-u.ac.jp; 4Division of Occupational Therapy, School of Health Sciences, Faculty of Medicine, Shinshu University, 3-1-1, Asahi, Matsumoto, Nagano 390-8621, Japan; sagaria@shinshu-u.ac.jp; 5Department of Occupational Therapy, Faculty of Rehabilitation, Kyushu Nutrition Welfare University, 1-5-1, Kuzuharatakamatsu, Kokuraminami-ku, Kitakyushu, Fukuoka 800-0298, Japan; 814.miya.418@gmail.com; 6Department of Neuropsychiatry, Kumamoto University Hospital, 1-1-1 Honjo Chuo-ku, Kumamoto 860-8556, Japan; hans11057@gmail.com (G.H.); kazuzak@kuh.kumamoto-u.ac.jp (K.Y.); 7Graduate Course of Health and Social Services, Saitama Prefectural University, 820, Sannomiya, Koshigaya, Saitama 343-8540, Japan; kawagoe-masahiro@spu.ac.jp

**Keywords:** Long-term care insurance, geriatric, healthcare, outpatient rehabilitation, outpatient day long-term care

## Abstract

Outpatient rehabilitation (OR) and outpatient day long-term care (ODLC) services are frequently used by older adult patients in Japan. However, there is a need to clarify that OR service, which has more rehabilitation professionals than ODLC, has the role of providing rehabilitation. This retrospective study examined the impact of OR services by comparing the two services based on City A data from the care-needs certification survey conducted between 2015 to 2017. We performed a propensity score matched analysis to compare the changes in the care level and function of OR and ODLC users after two years. The results showed that OR users showed a lower deterioration in care levels and less decline in the activities of daily living (ADL) in dementia and adaptation to social life. In the analysis of older adults requiring support, OR users had a lower deterioration in care levels and less decline in the ADL in dementia and behavioral and psychological symptoms than ODLC users did. There was no difference between the two services with respect to older adults requiring long-term care. The OR service has had an increasingly preventive effect on the deterioration of care levels compared to the ODLC service, which was particularly evident in older adults requiring support.

## 1. Introduction

Many developed countries face the problem of population aging, as the proportion of older adults requiring care has increased globally [1]. Japan has become a super-aging society due to its high life expectancy and low birth rate [2,3]. The proportion of older adults (aged ≥ 65 years) in Japan was the highest worldwide in 2017 at 27.7%, which is predicted to increase to about 40% by 2055 [4]. Geriatric syndromes refer to multi-etiological disorders associated with physiological aging, thus representing a major problem in aging populations. Frailty, falls, cognitive impairment, and urinary incontinence are the most common geriatric syndromes [5]. Under the Japanese long-term care insurance (LTCI), many issues requiring nursing care are associated with geriatric syndromes, including dementia (18.7%), fall/fractures (12.5%), and frailty (13.8%) [4].

In April 2000, the Japanese government implemented the LTCI system to address accelerated aging and the increased need for care for older adults [3,6,7]. The LTCI system emphasizes long-term care prevention and primarily aims to enable the older adults to live independently in their communities for as long as possible, to improve their health condition, and to prevent deterioration so as to preclude the need for long-term care [8,9]. The LTCI certification system is a two-stage process, which categorizes individuals into seven levels: support levels 1 and 2 and care needs levels 1 to 5. In the first stage, based on a certification survey, the required duration of long-term care is estimated using a computer program. In the second stage, a committee of physicians and other health care professionals determine the ranking. Individuals who qualify for long-term care can receive benefits, with the services divided into two categories: long-term care benefits (care needs levels 1 to 5) and prevention benefits (support levels 1 and 2). 

The number of individuals who qualify for long-term care has increased nearly threefold from 2.18 million in 2000 to 6.08 million in 2015 [10]. In addition, service use has increased, with 5.11 million people (of which, 3.82 million use in-home services) receiving benefits in 2015 [10]. Daycare services are one of the commuting services used by older adults at home. Although daycare models vary across different countries and have different insurance systems, many older adults all over the world use these services [11,12,13]. The utilization of daycare has been reported to be effective in preventing cognitive function and disability decline [13], reducing caregiver burden [14,15,16,17], as well as preventing the impairment of life functions in individuals with dementia [18], and behavioral and psychological symptoms of dementia (BPSD) [16,17,19,20,21]. On the other hand, an international systematic review found that the effectiveness of daycare is difficult to assess because of the lack of a standardized definition of daycare [22]. In Japan, commuting services such as outpatient rehabilitation (OR) and outpatient day long-term care (ODLC) are available for older adults living at home to use and receive rehabilitation and physical exercises from. In the OR service, patients visit a health facility to undergo training for sustaining or recovering their mental or physical functions and for leading an independent daily life. In the ODLC service, patients visit a health facility to receive care including baths, meals, recovery therapy, and recreation. The basic policy of OR is “to maintain and recover the physical and mental functions of users using physical therapy, occupational therapy, and other necessary rehabilitation [23].” Meanwhile, the basic policy of ODLC is “to eliminate user social isolation, maintain user physical and mental functions, and reduce the burden on families by supporting activities of daily living (ADL) and physical exercises [24].” These two services are similar in terms of “aiming to maintain or improve daily life function so that they can live independent lives according to the capabilities of users [23,24].” 

Recently, both these services have been required to clarify their roles given the similarity in the services provided to the users. According to a prior survey conducted by the Ministry of Health, Labour and Welfare (MHLW), the proportions of rehabilitation professionals, including physical therapists (PT), occupational therapists (OT), and speech-language-hearing-therapist (ST), between these services were significantly different, and OR had a higher number of rehabilitation professionals [25]. Previous studies have shown the effectiveness of rehabilitation, including PT and OT, in home care service [26,27]. The OR service, which has a higher number of rehabilitation professionals, is expected to provide high-quality rehabilitation services for the purpose of preventing the need for care. It should be made clear that the role of OR is to provide rehabilitation. According to the MHLW survey, OR was reported to improve independent ADL compared to ODLC [25]. However, the survey did not adequately control for service users’ factors that could have influenced the results. Physical function [28,29], cognitive function [30], dementia [31,32,33], frailty [34,35], age [36], and use of services [36,37,38,39] have been identified as factors associated with long-term care certification and changes in the care level. Since many factors are related to the level of care needed, it is necessary to consider these factors to investigate the impact of LTCI services. This study aimed to examine the effect of OR services by comparing it with ODLC services, by performing a propensity score matched analysis.

## 2. Materials and Methods 

### 2.1. Study Population and Data Sources

This retrospective study used Japanese LTCI certification investigation data. The study examined the LTCI certification investigation data of City A from 2015 to 2017. A total of 18,929 participants were observed between 2015 and 2017. Of them, 1642 participants (mean age 82.5 ± 6.7 years, female 70.8%) who continued using the OR (591 individuals) or ODLC services (1051 individuals) for over 2 years since 2015 were analyzed (Figure 1). They did not use other LTCI services. 

The National Institute of Population and Social Security Research (with which Masahiro Kawagoe was previously affiliated) signed a memorandum of understanding with City A concerning the use of data, which was approved by the ethics committee (IPSS-TRN#15001-2). Owing to a change in affiliation for Kawagoe, Saitama Prefectural University signed a new memorandum of understanding with City A.

### 2.2. Measurement

We analyzed data based on the certification of the long-term care needs survey. The certification survey item consists of 74 items, including 62 items on physical and mental health and 12 items on special medical care, and it is divided into the following five domains: body function/bed mobility, daily life function, cognitive function, behavioral and psychological symptoms, and adaptation to social life. In this study, total points were calculated for each of the five domains of the certification survey items and investigated. Additionally, we investigated the degree of independent daily living for disabled older adults and the degree of independent daily living for older adults with dementia

#### 2.2.1. Body Function/Bed Mobility

Body function/Bed mobility consists of the following 13 items: “paralysis,” “joint contracture,” “rolling,” “sitting up,” “sitting,” “standing,” “walking,” “stand-up,” “single-leg standing,” “washing the body,” “cutting nails,” “eyesight,” and “hearing ability.” “Paralysis” is scored from 0 to 5 points based on the presence or absence of limb paralysis and the presence or absence of the lost limbs. “Contracture” is scored from 0 to 4 points depending on the presence or absence of the restriction of the shoulder joint, hip joint, knee joint, and the other joints’ range of motion. By the evaluation of ability, “rolling,” “sitting up,” “standing,” “walking,” “stand-up,” and “single-leg standing” are scored from 1 to 3, “sitting” is scored from 1 to 4, and “eyesight” and “hearing ability” are scored from 1 to 5. Depending on the degree of assistance,” “washing the body” is scored from 1 to 4, and “cutting nails” is scored from 1 to 3. Higher scores indicate a lower body functioning/bed mobility and an increased need for assistance.

#### 2.2.2. Daily Life Function

Daily life function consists of the following 12 items: “transfer,” “mobility,” “swallowing,” “eating,” “toilet hygiene (urinary),” “toilet hygiene (fecal),” “brushing teeth/rinsing mouth,” “washing face,” “combing/styling hair,” “upper-body dressing,” “lower-body dressing,” and “frequency of going out.” Depending on the degree of assistance, “transfer,” “mobility,” “eating,” “toilet hygiene (urinary),” “toilet hygiene (fecal),” “upper-body dressing,” and “lower-body dressing” are scored from 1 to 4, and “brushing teeth/rinsing mouth,” “washing face,” and “combing/styling hair” are scored from 1 to 3. By the evaluation of ability, “swallowing” is scored from 1 to 3. The item of “frequency of going out” is scored from 1 to 3 depending on the frequency of going out. Higher scores indicate a lower daily life functioning and an increased need for assistance.

#### 2.2.3. Cognitive Function

Cognitive function consists of the following 9 items: “communicate intentions to others,” “understanding of daily routine,” “remembering date of birth,” “short-term memory,” “remembering own name,” “recognizing the season of the year,” “location awareness,” “wandering,” and “being unable to return to own room or home.” The evaluation of the ability to “communicate intentions to others” is scored from 1 to 4, and “understanding of daily routine,” “remembering date of birth,” “short-term memory,” “remembering own name,” “recognizing the season of year,” and “location awareness” are scored from 1 to 2. Depending on the presence or absence of symptoms and their frequency, “wandering” and “being unable to return to own room or home” are scored from 1 to 3. Higher scores indicate a lower cognitive ability.

#### 2.2.4. Behavioral and Psychological Symptoms

Behavioral and psychological disorders consist of the following 15 items: “paranoid behavior,” “confabulation,” “emotional instability,” “day-night reversal,” “repetitive talk,” “screaming,” “resistance to care,” “restlessness due to request to go home,” “behavior to go out alone,” “collectionism,” “breaks things or tearing off clothing,” “forgetfulness,” “talking to oneself/ laughing by oneself,” “selfish behavior inappropriate for the situation,” and “incoherent talk.” Depending on the presence or absence of symptoms and their frequency, all items are scored from 1 to 3. Higher scores indicate a higher prevalence of behavioral and psychological symptoms.

#### 2.2.5. Adaptation to Social Life

Adaptation to social life consists of the following 6 items: “own medication,” “handling finances,” “daily decision making,” “maladaptation to a group,” “shopping,” and “simple cooking.” Depending on the degree of assistance, “own medications” and “handling finances” are scored from 1 to 3, and “shopping” and “simple cooking” are scored from 1 to 4. The item of “daily decision making” is scored from 1 to 4 depending on the ability. The item of “maladaptation to a group” is scored from 1 to 4 depending on the presence or absence of symptoms and their frequency. Higher scores indicate a lower adaptation to social life and an increased need for assistance.

#### 2.2.6. Degree of Independent Daily Living for Disabled Older Adults 

This index designated by the MHLW indicates the level of independence in daily living in older adults with disabilities, and the severity is graded as follows: independent, J1, J2, A1, A2, B1, B2, C1, and C2 (Appendix A). Level J is defined as “although the patient has some disability, their daily living is almost independent and they go out on their own (J1: the patients can go out by using public transportation, J2: the patient can go out in the neighborhood),” Level A is defined as “the patients can live almost independently at home, but they cannot go out without a caregiver (A1: the patient goes out with assistance and lives mostly away from the bedside during the daytime, A2: the patient has little frequency in going out, and they live sleeping or getting up during the daytime),” Level B is defined as “the patients requires some care in their living at home, and mainly live on the bed during the daytime (B1: the patient can transfer to the wheelchair and eat and excrete away from the bed, B2: the patient transfers to the wheelchair with assistance),” and Level C is defined as “the patient is bedridden and requires care for excretion, eating, and dressing (C1: the patient can roll over, C2: the patient cannot roll over).”

#### 2.2.7. Degree of Independent Daily Living for Older Adults with Dementia

This index, also designated by the MHLW, indicates the level of independence in daily living in older adults with dementia, and the severity is graded as follows: independent, I, IIa, IIb, IIIa, IIIb, IV, and M (Appendix A). Level I is defined as “the patient has some dementia, but can live independently at home and in society,” Level II is defined as “although the patient has some symptoms or behaviors that disturb daily living, they can live independently with the attention and support of others (IIa: symptoms or behavior are present outside the home, IIb: symptoms or behavior are present in the home),” Level III is defined as “the patient has symptoms or behaviors that disturb daily living, and they require care (IIIa: symptoms or behaviors mainly present in the daytime, IIIb: symptoms or behaviors mainly present in the nighttime),” Level IV is defined as “the patient frequently has symptoms or behaviors that disturb daily living, and they always require care,” and Level M is defined as “the patient has severe mental symptoms, behavioral and psychological symptoms of dementia, or a severe physical disease, and therefore requires special treatment.” 

### 2.3. Propensity Score Matching

We employed propensity score matching to balance the effect of potential confounding biases in the comparison analyses between the OR service users and ODLC service users [40,41]. We used a logistic regression of the following factors for the propensity score calculation: age, sex, level of care needed, the degree of independent daily living for disabled older adults, the degree of independent daily living for older adults with dementia, body function/bed mobility, daily life function, cognitive function, behavioral and psychological symptoms, and adaptation to social life. The above factors used to calculate the propensity score were from 2015 onwards. One to one matching without a replacement was performed. We selected a caliper distance of 0.20 of the standard deviation of the logit of the propensity score.

### 2.4. Outcomes

We compared the certification survey item scores between 2015 and 2017 for each of the two groups after the propensity matching. After that, based on the certification survey item scores in 2015 and 2017, the variation was calculated and compared between the two groups. The rate of maintenance/improvement and deterioration was calculated for the level of care needed, the degree of independent daily living for disabled older adults, and the degree of independent daily living for older adults with dementia; these were compared between the two groups.

### 2.5. Statistical Analysis

Statistical calculations were performed using IBM SPSS Statistics version 24.0 (IBM Corp., Armonk, NY, USA). Student t-tests and a Pearson’s χ^2^ test were performed to examine the differences in the characteristics before and after the propensity score matching. The certification survey item scores between 2015 and 2017 were compared using paired t-tests. Student t-tests were performed to compare the variation in the certification survey items, and the Pearson’s χ^2^ test was performed to compare changes in the level of care needed, the degree of independent daily living for disabled older adults, and the degree of independent daily living for older adults with dementia.

## 3. Results

### 3.1. Sample Characteristics Before and After Propensity Score Matching

The characteristics of the entire sample before and after the propensity score matching are represented in Table 1. Significant differences were observed between the sex (*p* < 0.001), body function/bed mobility (*p* = 0.007), cognitive function (*p* = 0.010), behavioral and psychological symptoms (*p* < 0.001), level of care needed (*p* < 0.001), and the degree of independent daily living for older adults with dementia (*p* < 0.001) before the propensity score matching. The propensity score matching created 580 matched pairs (C-statistic: 0.632), and there were no significant differences in any factors among the two groups.

The characteristics associated with the need for support before and after the propensity score matching are presented in Table 2. Significant differences were observed between the sex (*p* < 0.001), behavioral and psychological symptoms (*p* = 0.017), and adaptation to social life (*p* = 0.043) before the propensity score matching. The propensity score matching created 289 matched pairs (C-statistic: 0.620), and there were no significant differences in any factors between the groups.

The characteristics associated with the need for long-term care before and after the propensity score matching are represented in Table 3. There were significant differences between the sex (*p* < 0.001), body function/bed mobility (*p* = 0.003), daily life function (*p* = 0.013), behavioral and psychological symptoms (*p* = 0.018), level of care needed (*p* = 0.019), and the degree of independent daily living for older adults with dementia (*p* < 0.001) before the propensity score matching. The propensity score matching created 271 matched pairs (C-statistic: 0.627), and there were no significant differences in any of the factors between the groups.

### 3.2. Changes in Certification Survey Item Scores after 2 Years

In all of the analyzed groups, both the OR and ODLC service users reported significantly declined functions for all of the domains (Table 4).

### 3.3. Comparisons of the Certification Survey Items after Two Years Between Outpatient OR and ODLC Service Users

The results of the comparisons between the groups on the variation of the certification survey items are represented in Table 5. In the overall sample, the OR service users had a maintained or significantly improved level of care needed (*p* = 0.001) and degree of independent daily living for older adults with dementia (*p* = 0.006), compared to the ODLC service users. Furthermore, the OR service users showed significantly less decline in the adaptation to social life compared to the ODLC service users (*p* = 0.029). Regarding the need for support, the OR service users reported significantly less changes in behavioral and psychological symptoms than did the ODLC service users (*p* = 0.022). In addition, the OR service users had a maintained or significantly improved level of care needed (*p* < 0.001) and degree of independent daily living for older adults with dementia (*p* < 0.001). There were no significant differences in the need for long-term care.

## 4. Discussion

This study showed that, compared to ODLC, OR is more effective at preventing the deterioration of the care level. This effect was particularly evident when support was needed. In addition, compared to ODLC, OR prevented the deterioration of ADL in dementia, in the adaptation to social life and in those patients requiring support. However, there was no difference in the preventive effect of long-term care between both services regarding the need for long-term care.

In a previous study of changes in care levels after two years, the care levels of half of the patients requiring support and 30–40% of patients requiring long-term care deteriorated [38]. In the present study, the proportions of patients requiring support were 33.9% of the OR service users and 52.2% of the ODLC service users with deteriorated care levels. Particularly, OR service users in this study had less deterioration of the care level than those in previous studies did. As in previous studies that required long-term care, about 30% of both service users had deteriorated care levels. 

According to MHLW, compared to ODLC service, the OR service has numerous contacts to rehabilitation professionals, such as PT, OT, and ST (in OR, PT is 1.7, OT is 0.9, and ST is 0.2 per office, while in ODLC, PT is 0.12, OT is 0.06, and ST is 0.01 per office), so that the rate of utilization of ADL evaluation indicators is high and cooperation with medical doctors is established [25]. It is considered effective in preventing the deterioration of care levels and ADL by an intervention based on the objective assessment of ADL by rehabilitation professionals. The physical and mental functions, including the body, daily life, and cognitive functions, were less impaired in all domains for the OR users than for the ODLC users, although this was not significant. Rehabilitation focusing on activity and participation based on the International Classification of Disability, Health and Functioning (ICF) is suggested to improve ADL independence. These are considered to be factors that show significant differences, especially in the care level and ADL. Furthermore, in OR, efforts are being made that involve not only rehabilitation professionals, but also multidisciplinary professionals including medical doctors, nurses, carers, and care managers, with the aim of improving the daily life performance for older adults that require long-term care. We consider that a multidisciplinary approach will be effective in preventing the deterioration of the care level and ADL.

In patients requiring support, our results indicated that a decline in ADL and BPSD is one of the small factors that have a preventive effect on the deterioration of care levels in OR. Dementia is associated with changes in the care level [31,32,33], and the degree of independence in ADL in older adults with dementia in the certification survey represents the degree of necessity of care due to a decline in the cognitive function and BPSD. The degree of independence in ADL for the older adults with dementia in requiring support was 90% of individuals at the independent level or mild level I. Rehabilitation for people with dementia is effective, especially for mild to moderate dementia [42,43,44], and it may have been effective in those requiring support who are mostly occupied by mild dementia. The domain of behavioral and psychological symptoms investigated in this study are items related to BPSD [45], and compared with the ODLC service, the OR service had prevented the deterioration of BPSD. BPSD is associated with a high caregiver burden [46,47], and the use of care services is considered effective in reducing the BPSD and caregiver burden [16,17,19,20]. The use of the adult day-care service is reported to improve night-time sleep-related problems and depressive behaviors [17,20,21]. Commuting services including OR and ODLC are effective in reducing apathy [48], the most frequent symptom of BPSD [49,50]. In OR, it is possible that BPSD has been dealt with more appropriately, as rehabilitation professionals intervene and there is cooperation with medical doctors. BPSD is a major contributor to caregiver burden and early institutionalization [51]; thus, preventing the deterioration of BPSD is especially important from the perspective of the user and their families and also from the perspective of continuing home-living. However, various factors are related to changes in the care level [26,27,28,29,30,31,32,33,34,35,36,37], which cannot be specified based on the results of this study.

In previous studies, the effect of each LTCI service on the eligibility level was often reported for individuals requiring light long-term care and support [52,53,54]. The effect of OR compared to ODLC was limited to requiring support even in the present study. The OR and ODLC also provide services, including meals, transportation, bathing, and easy recreation by care workers. According to previous surveys by the MHLW, the individual rehabilitation duration is as short as 16.6 min for OR and 22.4 min for ODLC, suggesting the possibility that sufficient benefits have not been obtained for those requiring long-term care whose functionality is affected by advanced deterioration. For the OR service, improving the rehabilitation function and reducing the severity of care levels are needed. Individual rehabilitation time is limited; thus, it is necessary to efficiently accommodate those who require long-term care with advanced functional deterioration.

Our findings support the policy for a functional differentiation of OR and ODLC by the MHLW, which is currently underway. It should be made clear that the role of OR is to provide rehabilitation. The OR service should provide a high-quality rehabilitation by promoting the placement of rehabilitation professionals and enhancing the services. In particular, it is necessary to improve the ADL of older adults who require care. The provision of the OR and ODLC services should be tailored to the needs of the users and their families. As daycare service models vary internationally, it will be difficult to adapt our results directly into the systems of other countries. However, promoting the placement of rehabilitation professionals in commuting services and ensuring coordination with them will help improve the care-need level of the recipient. In addition, having a clear understanding of the role of each service helps the users and their families to use the service efficiently.

Some study limitations should be considered. First, as our data were obtained from individuals certified with LTCI among older adults living in a single city, the generalizability of the sample is limited. Future replications studies in other cities or prefectures are needed to confirm our results. Second, although we used propensity score matching to control for the user factors obtained from the certification survey data, it is not clear which causative diseases require the most care. The causative disease is a major factor affecting long-term care. Third, detailed data on the use of services is not sufficient. The frequency of the service use [37] and the content of the service are also important factors related to long-term care, although this study did not investigate these factors. Moreover, family composition and the presence or absence of social support are related to declining functions [55,56], and are considered to influence the use of care services. More detailed personal data on the use of these services are needed to clarify the effectiveness of the OR service. Finally, we did not study the effectiveness of OR by comparing OR and a control group; instead, we compared two specific services. Therefore, the relevance of the results is limited. Future studies are necessary to accumulate sufficient evidence on the effect of the OR service. Despite these limitations, the present study shows important results, which can be used as basis to improve the quality of OR services and thereby improve the care levels of older adult patients with long-term care certification in future LTCI.

## 5. Conclusions

Our findings suggest that compared to the ODLC service, the OR service has a preventive effect on the deterioration of the care levels for older adults requiring support. Promoting the placement of rehabilitation professionals in commuting services and ensuring the collaboration with multidisciplinary professionals will help improve the care-need level of the recipient. Additionally, having a clear understanding of the role of services related to the older adults requiring long-term care—including a long-term care insurance—helps the service users and their families to use the service efficiently. Because there were no differences in the effectiveness of both services in older adults requiring long-term care, the OR service requires efficient efforts for those that require long-term care and for those who are more functionally deteriorated.

## Figures and Tables

**Figure 1 ijerph-16-01292-f001:**
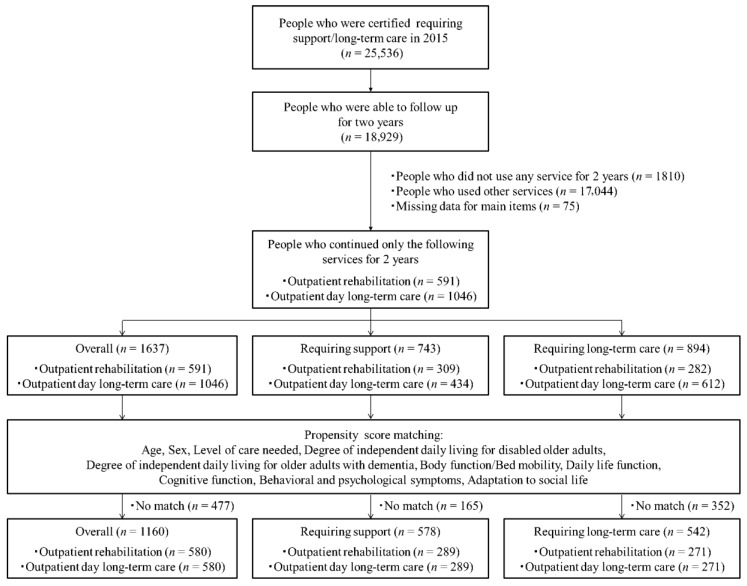
Flowchart of inclusion and exclusion criteria for this study.

**Table 1 ijerph-16-01292-t001:** Sample characteristics before and after propensity score matching.

	Before Matching			After Matching		
	OR Service User (*n* = 591)	ODLC Service User (*n* = 1046)	*p*-Value	OR Service User (*n* = 580)	ODLC Service User (*n* = 580)	*p*-Value
Age, years, mean (SD)	82.1(7.0)	82.7(6.5)	0.086	82.1(7.0)	82.2(6.6)	0.796
Female sex, *n*(%)	376(63.6)	782(74.8)	0.000	375(64.7)	385(66.4)	0.537
Body function/Bed mobility, mean (SD)	19.6(3.9)	19.0(4.0)	0.007	19.5(3.8)	19.3(4.0)	0.430
Daily life function, mean (SD)	14.3(3.8)	14.1(3.4)	0.440	14.2(3.5)	14.1(3.4)	0.819
Cognitive function, mean (SD)	9.6(1.2)	9.7(1.2)	0.010	9.6(1.2)	9.6(1.2)	0.820
Behavioral and psychological symptoms, mean (SD)	17.8(3.4)	18.7(3.7)	0.000	17.8(3.4)	17.9(3.3)	0.535
Adaptation to social life, mean (SD)	12.4(3.5)	12.5(3.5)	0.509	12.4(3.5)	12.2(3.6)	0.345
Level of care needed, *n* (%)			0.000			0.402
Requiring support 1	134(22.7)	213(20.4)		133(22.9)	142(24.5)	
Requiring support 2	175(29.6)	221(21.1)		171(29.5)	162(27.9)	
Requiring long-term care 1	164(27.7)	405(38.7)		163(28.1)	177(30.5)	
Requiring long-term care 2	100(16.9)	165(15.8)		98(16.9)	83(14.3)	
Requiring long-term care 3	13(2.2)	37(3.5)		13(2.2)	11(1.9)	
Requiring long-term care 4	3(0.5)	5(0.5)		1(0.2)	5(0.9)	
Requiring long-term care 5	2(0.3)	0(0)		1(0.2)	0(0)	
Degree of independent daily living for disabled older adults, *n* (%)			0.434			0.995
J1	3(0.5)	10(1.0)		3(0.5)	4(0.7)	
J2	198(33.5)	356(34.0)		198(34.1)	200(34.5)	
A1	197(33.3)	377(36.0)		194(33.4)	197(34.0)	
A2	167(28.3)	266(25.4)		163(28.1)	156(26.9)	
B1	18(3.0)	30(2.9)		17(2.9)	17(2.9)	
B2	7(1.2)	7(0.7)		5(0.9)	6(1.0)	
C1	1(0.2)	0(0)		0(0)	0(0)	
C2	0(0)	0(0)		0(0)	0(0)	
Degree of independent daily living for older adults with dementia, *n* (%)			0.000			0.996
Indepedence	144(24.4)	200(19.1)		140(24.1)	137(23.6)	
I	190(32.1)	236(22.6)		185(31.9)	185(31.9)	
IIa	61(10.3)	122(11.7)		61(10.5)	60(10.3)	
IIb	133(22.5)	354(33.8)		132(22.8)	137(23.6)	
IIIa	55(9.3)	115(11.0)		55(9.5)	56(9.7)	
IIIb	6(1.0)	14(1.3)		5(0.9)	3(0.5)	
IV	2(0.3)	5(0.5)		2(0.3)	2(0.3)	
M	0(0)	0(0)		0(0)	0(0)	

SD, standard deviation; OR, outpatient rehabilitation; ODLC, outpatient day long-term care; The p value was calculated using the χ^2^ test for the categorical data; The p value was calculated using the student t-tests for continuous measures.

**Table 2 ijerph-16-01292-t002:** Sample characteristics before and after propensity score matching in older adults requiring support.

	Before Matching			After Matching		
	OR Service User (*n* = 309)	ODLC Service User (*n* = 434)	*p*-Value	OR Service User (*n* = 289)	ODLC Service User (*n* = 289)	*p*-Value
Age, years, mean (SD)	81.9(6.9)	82.2(6.2)	0.580	82.0(6.9)	82.1(6.4)	0.807
Female sex, *n* (%)	213(68.9)	351(80.9)	0.000	208(72.0)	214(74.0)	0.574
Body function/Bed mobility, mean (SD)	18.8(3.1)	18.4(3.0)	0.078	18.7(3.0)	18.7(3.0)	0.764
Daily life function, mean (SD)	12.7(1.3)	12.6(1.2)	0.461	12.7(1.3)	12.7(1.2)	0.920
Cognitive function, mean (SD)	9.1(0.3)	9.1(0.2)	0.594	9.0(0.2)	9.0(0.2)	1.000
Behavioral and psychological symptoms, mean (SD)	15.9(1.6)	16.2(1.9)	0.017	16.0(1.6)	16.0(1.5)	0.895
Adaptation to social life, mean (SD)	10.6(3.3)	10.1(3.2)	0.043	10.4(3.2)	10.2(3.2)	0.503
Level of care needed, n (%)			0.124			0.182
Requiring support 1	134(43.4)	213(49.1)		127(43.9)	143(49.5)	
Requiring support 2	175(56.6)	221(50.9)		162(56.1)	146(50.5)	
Degree of independent daily living for disabled older adults, *n* (%)			0.253			0.989
J1	2(0.6)	8(1.8)		2(0.7)	3(1.0)	
J2	147(47.6)	211(48.6)		139(48.1)	141(48.8)	
A1	92(29.8)	143(32.9)		89(30.8)	85(29.4)	
A2	65(21.0)	69(15.9)		58(20.1)	59(20.4)	
B1	3(1.0)	2(0.5)		1(0.3)	1(0.3)	
B2	0(0)	1(0.2)		0(0)	0(0)	
C1	0(0)	0(0)		0(0)	0(0)	
C2	0(0)	0(0)		0(0)	0(0)	
Degree of independent daily living for older adults with dementia, *n* (%)			0.292			0.928
Indepedence	131(42.4)	193(44.5)		128(44.3)	128(44.3)	
I	144(46.6)	182(41.9)		131(45.3)	133(46.0)	
IIa	23(7.4)	29(6.7)		19(6.6)	19(6.6)	
IIb	10(3.2)	28(6.5)		10(3.5)	7(2.4)	
IIIa	1(0.3)	2(0.5)		1(0.3)	2(0.7)	
IIIb	0(0)	0(0)		0(0)	0(0)	
IV	0(0)	0(0)		0(0)	0(0)	
M	0(0)	0(0)		0(0)	0(0)	

SD, standard deviation; OR, outpatient rehabilitation; ODLC, outpatient day long-term care; The *p*-value was calculated using the χ^2^ test for the categorical data; The *p*-value was calculated using the student *t*-tests for continuous measures.

**Table 3 ijerph-16-01292-t003:** Sample characteristics before and after propensity score matching in older adults requiring long-term care.

	Before Matching			After Matching		
	OR Service User (*n* = 282)	ODLC Service User (*n* = 612)	*p*-Value	OR Service User (*n* = 271)	ODLC Service User (*n* = 271)	*p*-Value
Age, years, mean (SD)	82.3(7.0)	83.1(6.7)	0.119	82.4(7.0)	82.6(7.3)	0.695
Female sex, *n* (%)	163(57.8)	431(70.4)	0.000	161(59.4)	166(61.3)	0.661
Body function/Bed mobility, mean (SD)	20.4(4.5)	19.5(4.6)	0.003	20.2(4.3)	20.5(4.7)	0.416
Daily life function, mean (SD)	16.0(4.7)	15.2(4.0)	0.013	15.8(4.4)	15.6(4.2)	0.542
Cognitive function, mean (SD)	10.1(1.5)	10.2(1.4)	0.434	10.2(1.5)	10.1(1.4)	0.479
Behavioral and psychological symptoms, mean (SD)	19.8(3.7)	20.4(3.7)	0.018	19.9(3.7)	19.9(3.5)	0.914
Adaptation to social life, mean (SD)	14.4(2.3)	14.3(2.6)	0.356	14.4(2.3)	14.2(2.6)	0.297
Level of care needed, n (%)			0.019			0.706
Requiring long-term care 1	164(58.2)	405(66.2)		162(59.8)	165(60.9)	
Requiring long-term care 2	100(35.5)	165(27.0)		94(34.7)	87(32.1)	
Requiring long-term care 3	13(4.6)	37(6.0)		12(4.4)	17(6.3)	
Requiring long-term care 4	3(1.1)	5(0.8)		2(0.7)	2(0.7)	
Requiring long-term care 5	2(0.7)	0(0)		1(0.4)	0(0)	
Degree of independent daily living for disabled older adults, *n* (%)			0.167			0.947
J1	1(0.4)	2(0.3)		1(0.4)	2(0.7)	
J2	51(18.1)	145(23.7)		50(18.5)	49(18.1)	
A1	105(37.2)	234(38.2)		102(37.6)	96(35.4)	
A2	102(36.2)	197(32.2)		99(36.5)	102(37.6)	
B1	15(5.3)	28(4.6)		14(5.2)	18(6.6)	
B2	7(2.5)	6(1.0)		5(1.8)	4(1.5)	
C1	1(0.4)	0(0)		0(0)	0(0)	
C2	0(0)	0(0)		0(0)	0(0)	
Degree of independent daily living for older adults with dementia, *n* (%)			0.000			0.996
Indepedence	13(4.6)	7(1.1)		7(2.6)	7(2.6)	
I	46(16.3)	54(8.8)		42(15.5)	38(14.0)	
IIa	38(13.5)	93(15.2)		38(14.0)	40(14.8)	
IIb	123(43.6)	326(53.3)		123(45.4)	125(46.1)	
IIIa	54(19.1)	113(18.5)		54(19.9)	55(20.3)	
IIIb	6(2.1)	14(2.3)		5(1.8)	5(1.8)	
IV	2(0.7)	5(0.8)		2(0.7)	1(0.4)	
M	0(0)	0(0)		0(0)	0(0)	

SD, standard deviation; OR, outpatient rehabilitation; ODLC, outpatient day long-term care; The *p*-value was calculated using the χ^2^ test for the categorical data; The *p*-value was calculated using the student *t*-tests for continuous measures.

**Table 4 ijerph-16-01292-t004:** Comparison of certification care-needs survey items between 2015 and 2017.

	Overall						Requiring Support						Requiring Long-Term Care					
	OR Service User (*n* = 580)			ODLC Service User (*n* = 580)			OR Service User (*n* = 289)			ODLC Service User (*n* = 289)			OR Service User (*n* = 271)			ODLC Service User (*n* = 271)		
	2015	2017	*p*-Value	2015	2017	*p*-Value	2015	2017	*p*-Value	2015	2017	*p*-Value	2015	2017	*p*-Value	2015	2017	*p*-Value
Body function/Bed mobility, mean (SD)	19.5(3.8)	20.6(4.6)	0.000	19.3(4.0)	20.5(4.6)	0.000	18.7(3.0)	19.8(4.1)	0.000	18.7(3.0)	20.4(4.4)	0.000	20.2(4.3)	21.2(4.8)	0.000	20.5(4.7)	21.5(5.2)	0.000
Daily life function, mean (SD)	14.2(3.5)	15.6(5.8)	0.000	14.1(3.4)	15.7(5.7)	0.000	12.7(1.3)	14.0(5.0)	0.000	12.7(1.2)	14.4(5.4)	0.000	15.8(4.4)	17.2(6.1)	0.000	15.6(4.2)	17.5(6.5)	0.000
Cognitive function, mean (SD)	9.6(1.2)	9.9(1.6)	0.000	9.6(1.2)	9.9(1.5)	0.000	9.0(0.2)	9.2(0.7)	0.000	9.0(0.2)	9.3(0.9)	0.000	10.2(1.5)	10.6(2.0)	0.000	10.1(1.4)	10.6(1.8)	0.000
Behavioral and psychological symptoms, mean (SD)	17.8(3.4)	18.3(4.0)	0.000	17.9(3.3)	18.7(4.0)	0.000	16.0(1.6)	16.6(2.6)	0.000	16.0(1.5)	17.1(3.0)	0.000	19.9(3.7)	20.3(4.3)	0.039	19.9(3.5)	20.5(4.1)	0.009
Adaptation to social life, mean (SD)	12.4(3.5)	13.0(3.5)	0.000	12.2(3.6)	13.1(3.6)	0.000	10.4(3.2)	11.1(3.5)	0.000	10.2(3.2)	11.3(3.6)	0.000	14.4(2.3)	15.0(2.2)	0.000	14.2(2.6)	15.0(2.5)	0.000

SD, standard deviation; OR, outpatient rehabilitation; ODLC, outpatient day long-term care; The *p*-value was calculated using the paired *t*-tests for continuous measures.

**Table 5 ijerph-16-01292-t005:** Comparisons of changes in the level of care and function after 2 years between OR and ODLC.

	Overall			Requiring Support			Requiring Long-Term Care		
	OR Service User (*n* = 580)	ODLC Service User (*n* = 580)	*p*-Value	OR Service User (*n* = 289)	ODLC Service User (*n* = 289)	*p*-Value	OR Service User (*n* = 271)	ODLC Service User (*n* = 271)	*p*-Value
Body function/Bed mobility, mean (SD)	−1.08(3.49)	−1.19(3.59)	0.613	−1.17(3.52)	−1.64(4.09)	0.133	−1.00(3.28)	−0.98(3.62)	0.941
Daily life function, mean (SD)	−1.40(5.40)	−1.56(4.99)	0.608	−1.29(5.07)	−1.72(5.20)	0.320	−1.39(5.32)	−1.98(5.65)	0.211
Cognitive function, mean (SD)	−0.31(1.10)	−0.36(1.12)	0.398	−0.17(0.73)	−0.28(0.86)	0.118	−0.44(1.34)	−0.55(1.36)	0.309
Behavioral and psychological symptoms, mean (SD)	−0.52(2.73)	−0.82(3.12)	0.084	−0.63(2.22)	−1.10(2.68)	0.022	−0.41(3.23)	−0.55(3.46)	0.608
Adaptation to social life, mean (SD)	−0.59(2.18)	−0.87(2.28)	0.029	−0.69(2.37)	−1.09(2.73)	0.060	−0.55(1.78)	−0.77(1.67)	0.143
Changes in level of care needed, improve/maintain, n(%)	402(69.3)	348(60.0)	0.001	191(66.1)	138(47.8)	0.000	197(72.7)	197(72.7)	1.000
Changes in degree of independent daily living for disabled older adults, improve/maintain, n(%)	458(79.0)	434(74.8)	0.095	229(79.2)	211(73.0)	0.079	213(78.6)	209(77.1)	0.679
Changes in degree of independent daily living for older adults with dementia, improve/maintain, n(%)	422(72.8)	379(65.3)	0.006	210(72.7)	168(58.1)	0.000	198(73.1)	194(71.6)	0.701

SD, standard deviation; OR, outpatient rehabilitation; ODLC, outpatient day long-term care; The *p*-value was calculated using the χ2 test for the categorical data; The *p*-value was calculated using the student *t*-tests for continuous measures.

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
