# Peer review of "Impact of Outpatient Rehabilitation Service in Preventing the Deterioration of the Care-Needs Level Among Japanese Older Adults Availing Long-Term Care Insurance: A Propensity Score Matched Retrospective Study"

_ijerph, 2019, doi:10.3390/ijerph16071292_

Round 1

Reviewer 1 Report

Accept after Minor Revisions

Maruta et al investigate the impact of OR service compared to ODLC service using propensity score matched analysis. OR service compared to ODLC service prevented deterioration of care level and ADL of people with dementia. The results were limited to older adults with support level. The authors conclude that the OR service has preventive effect on deteriorating care level compared with the ODLC service, especially in older adults with support level.

The current article would need to make some minor revisions from its current format, before it could progress to publication.

Materials and Methods

It is unclear what the total score of each measurement item indicates. According to the results section, show that the higher the total score is the lower the user's function?

Results

The authors stated that the certification survey item scores after two years had declined in the text, but Table 4 shows that the total score after two years is higher.

In Table 5, there is a significant difference in adaptation to social life among entire samples. It also needs to be described in the text.

Discussion

The authors considers that the fact that the OR service has more rehabilitation professionals than the ODLC service is a factor in preventing deterioration of care level. It is necessary to describe how many differences.

The contents of service provision will affect the results. The authors need to state them.

Author Response

Dear Reviewer 1

We appreciate your careful reading of our manuscript and your thoughtful suggestions to improve it. We have made revisions based on your comments and suggestions, as shown below. Your original comments are indicated in italics for reference, and our point-by-point responses are given below.

Materials and Methods

It is unclear what the total score of each measurement item indicates. According to the results section, show that the higher the total score is the lower the user's function?

Response:

Thank you for this comment. As pointed out, higher total scores represent lower user functionality. We added it to the Method section (p.4, line 134-135, 145, 155, 163, 171).

Results

The authors stated that the certification survey item scores after two years had declined in the text, but Table 4 shows that the total score after two years is higher.

Response:

We appreciate your comment. This is also relevant to your suggestion for the Method section. We revised “decreased scores” to “declined functions.” (p.9, line 255-256)

Results

In Table 5, there is a significant difference in adaptation to social life among entire samples. It also needs to be described in the text.

Response:

 We appreciate your comment, based on which we made a revision (p.11, line 266-267).

Discussion

The authors considers that the fact that the OR service has more rehabilitation professionals than the ODLC service is a factor in preventing deterioration of care level. It is necessary to describe how many differences.

Response:

 Thank you for pointing this out. We added the difference in the number of rehabilitation professionals between both groups reported by a previous study. In addition, we have added speech-language-hearing-therapist (ST) as a rehabilitation professional, because the description of ST was missing. (p.13, line 289-290).

Discussion

The contents of service provision will affect the results. The authors need to state them.

Response:

 Thank you for pointing this out. We agree that the impact of the contents of service provision is important in care prevention. However, the contents of the service were unknown in the data source we used. Therefore, we have added it as a limitation (p.14, line 346).

Reviewer 2 Report

The authors compared outpatient rehabilitation (OR) and outpatient day long-term care (ODLC) services in Japan. Propensity score matched analysis showed that OR is a superior service for elderly population in several aspects. The paper is well organized, methodologically sound, and the results are relevant even though the Japanese Ministry of Health, Labour and Welfare has already recognized inefficiencies of ODLC.

The only shortcoming of the paper is its language style. Even though the authors mentioned that the paper was edited, many sentences are difficult to follow since they are too long and complex.

Author Response

Dear Reviewer 2

We appreciate your careful reading of our manuscript and your useful comments. Your original comment is indicated in italics for reference, and our response is given below.

The authors compared outpatient rehabilitation (OR) and outpatient day long-term care (ODLC) services in Japan. Propensity score matched analysis showed that OR is a superior service for elderly population in several aspects. The paper is well organized, methodologically sound, and the results are relevant even though the Japanese Ministry of Health, Labour and Welfare has already recognized inefficiencies of ODLC.

The only shortcoming of the paper is its language style. Even though the authors mentioned that the paper was edited, many sentences are difficult to follow since they are too long and complex.

Response:

 We thank you for this comment. In accordance with your comment, we have revised the language style and received English language editing again.

Reviewer 3 Report

This is an interesting work that applies an useful method to overcome the limitations of observational data. The main problem with this manuscript is the interpretation of the results.After assessing the differences between OR and ODLC after 2, there are only significant differences obtained for certain of the different domains analyzed. As authors evaluate the effect in different outcome variables it is difficult to estimate what are the policy implications from this anlysis. My question here (not being an expert in this specific área) is which one of those domains could be considered from the perspective of older adults and their families more relevant.

This is highly relevant, given the differences in resources required for OR and ODLC, and considering that improving efficiency in those requiring long-term care is crucial.

Authors should clearly discuss how these results may be translated into practice.

Table 4 is not necessary, as the objective of this work is not to determine the diferences between 2015 and 2017 within the different alternatives evaluated.

Author Response

Dear Reviewer 3

We appreciate your careful reading of our manuscript and your thoughtful suggestions to improve it. We have made revisions based on your comments and suggestions, as shown below. Your original comments are indicated in italics for reference, and our point-by-point responses are given below.

This is an interesting work that applies an useful method to overcome the limitations of observational data. The main problem with this manuscript is the interpretation of the results.After assessing the differences between OR and ODLC after 2, there are only significant differences obtained for certain of the different domains analyzed. As authors evaluate the effect in different outcome variables it is difficult to estimate what are the policy implications from this anlysis. My question here (not being an expert in this specific área) is which one of those domains could be considered from the perspective of older adults and their families more relevant.

This is highly relevant, given the differences in resources required for OR and ODLC, and considering that improving efficiency in those requiring long-term care is crucial.

Response:

 Thank you for suggesting this important piece of information. We have made a revision based on your comment. We added that not all domains have significant results (p.13, line 293-298). We added information about the domains that are considered particularly important from the perspectives of user and their families (p.13, line 315-317).

Authors should clearly discuss how these results may be translated into practice.

Response:

 Thank you for pointing this out. We have revised this accordingly (p.14, line 329-339).

Table 4 is not necessary, as the objective of this work is not to determine the differences between 2015 and 2017 within the different alternatives evaluated.

Response:

 We appreciate your comment. As you pointed out, comparing the differences between 2015 and 2017 is not the main purpose of this study. However, we believe that comparing the preventive effects of OR and ODLC after showing the general trend of declining user's function in two years will help the reader's understanding. Therefore, we wish to retain Table 4.

Reviewer 4 Report

The study compares the impact of outpatient rehabilitation (OR) services to outpatient long-term care (ODLC) services in Japan. Propensity score matching is applied - for each OR user a 'statistical twin' is identified based on propensity scores among the ODLC users. In addition, the sample is split into people with low care needs (requiring support) and people with higher care needs (requiring long-term care). The outcome variables are used from a survey among service users, containing information on body function, daily life function, cognitive function, behavioral symptoms, adaption to social life. Changes between 2015 and 2017 are also observed.

I would suggest to make some revisions based on two points: 1) The study's methods and results are outlined in detail and seem to be of good quality, yet the study should consider to a larger extent studies beyond the Japanese context to be of relevance beyond the Japanese context. 2) The study compares to specific types of services - it should be made clear why these two groups (OR, ODLC) are compared instead of comparing one service with a group receiving none of these services (for example). This could provide more useful information for policymakers. 

ad 1) The study provides very detailed policy context for Japan, but fails to set the scene in a larger, global context. What is the relevance of the study for other countries? The policy context, while it is informative, could be shortened substantially (or a table could be added instead), and reference to the larger (global) context should be made (see page 2f.). Lines 76 to 93 on p.2 are confusing and need to be revised/made more concise.

ad 2) The relevance of comparing two specific service user groups is limited. I would recommend that the discussion section does not list results but instead provides the larger context and external validity of the findings. It should be clearly stated in the limitations why a comparison with those without any rehabilitation services is not possible (if it is not). 

Other comments:

Abstract - Please rephrase "The OR service was expected to provide higher quality rehabilitation; thus..." - the underlying rationale is not clear - why do you expect higher quality rehabilitation? 

Second, the reference group should be clarified in the abstract (line 34) "...psychological symptoms THAN..."

Third, the last part of the abstract (line 36) is imprecise, as for a reader not familiar with the text the term "support level" is confusing. (The authors refer to the care levels in the Japanese system, as becomes clear lateron).

Section 2.2.6 - It would be helpful to have a table instead of text to present the different care löevels and criteria.

Section 2.2.7 - The term 'demented' should be replaced throughout the text -as far as I know, 'people with dementia' is more appropriate.

 Abbreviations - please make sure all abbreviations are explained sufficiently (see PT and OT in the discussion section, for example)

Author Response

Dear Reviewer 4

We appreciate your careful reading of our manuscript and your thoughtful suggestions to improve it. We have made revisions based on your comments and suggestions, as shown below. Your original comments are indicated in italics for reference, and our point-by-point responses are given below.

The study compares the impact of outpatient rehabilitation (OR) services to outpatient long-term care (ODLC) services in Japan. Propensity score matching is applied - for each OR user a 'statistical twin' is identified based on propensity scores among the ODLC users. In addition, the sample is split into people with low care needs (requiring support) and people with higher care needs (requiring long-term care). The outcome variables are used from a survey among service users, containing information on body function, daily life function, cognitive function, behavioral symptoms, adaption to social life. Changes between 2015 and 2017 are also observed.

I would suggest to make some revisions based on two points: 1) The study's methods and results are outlined in detail and seem to be of good quality, yet the study should consider to a larger extent studies beyond the Japanese context to be of relevance beyond the Japanese context. 2) The study compares to specific types of services - it should be made clear why these two groups (OR, ODLC) are compared instead of comparing one service with a group receiving none of these services (for example). This could provide more useful information for policymakers.

Response:

 Thank you for pointing these out. We have provided our responses to the two major problems that you pointed out.

ad 1) The study provides very detailed policy context for Japan, but fails to set the scene in a larger, global context. What is the relevance of the study for other countries? The policy context, while it is informative, could be shortened substantially (or a table could be added instead), and reference to the larger (global) context should be made (see page 2f.). Lines 76 to 93 on p.2 are confusing and need to be revised/made more concise.

Response:

Thank you for pointing this out. We cited many studies related to the Japanese context, and therefore, we have now cited previous studies of daycare service, which is similar to OR and ODLC, beyond the Japanese context about (p.2, line 64-72). We have added about relevance of the findings to other countries in the Discussion (p.14, line 329-339). However, the relevance of the results is limited, as long-term care insurance services may vary internationally. We have revised the part (previous manuscript; lines 76 to 93 on p.2) that you pointed out (p.2, line 84-93, p.3, lines 94-95, 98-100).

ad 2) The relevance of comparing two specific service user groups is limited. I would recommend that the discussion section does not list results but instead provides the larger context and external validity of the findings. It should be clearly stated in the limitations why a comparison with those without any rehabilitation services is not possible (if it is not).

Response:

Thank you for your suggestions. In Japan, the Ministry of Health, Labor and Welfare revised its policy on the background of the problem that the two services OR and ODLC are similar. The OR service, which places more rehabilitation professionals compared to ODLC, requires clarification regarding the role of rehabilitation. In this study, we aimed to examine the effect of OR service compared to ODLC service. Therefore, we did not set a control group (receiving none of services). However, as pointed out, the relevance of the results is limited, so we have added this point as a limitation of the study (p.14, line 350-352).

Other comments:

Abstract - Please rephrase "The OR service was expected to provide higher quality rehabilitation; thus..." - the underlying rationale is not clear - why do you expect higher quality rehabilitation?

Response:

Thank you for pointing this out. We have revised this based on your comment.

Second, the reference group should be clarified in the abstract (line 34) "...psychological symptoms THAN..."

Response:

Thank you for pointing this out. We have revised this based on your comment.

Third, the last part of the abstract (line 36) is imprecise, as for a reader not familiar with the text the term "support level" is confusing. (The authors refer to the care levels in the Japanese system, as becomes clear lateron).

Response:

We appreciate your comment on this point. We agree that it could be difficult for the readers to understand. Accordingly, we revised "support level" to "older adults requiring support" and "care level" to "older adults requiring long-term care" in the abstract.

Section 2.2.6 - It would be helpful to have a table instead of text to present the different care löevels and criteria.

Response:

We appreciate your comment on this point. We have provided a supplementary Table accordingly.

Section 2.2.7 - The term 'demented' should be replaced throughout the text -as far as I know, 'people with dementia' is more appropriate.

Response:

We appreciate your comment on this point, and we have revised this accordingly. In addition, as pointed out regarding section 2.2.6, we have also provided a supplementary Table.

Abbreviations - please make sure all abbreviations are explained sufficiently (see PT and OT in the discussion section, for example)

Response:

Thank you for your suggestion. We have explained all abbreviations used in the manuscript.

Round 2

Reviewer 3 Report

This paper has improved after the review.

Author Response

Dear Reviewer 3

 We appreciate the time and effort you have dedicated to providing insightful feedback concerning ways to strengthen and improve our manuscript.

Reviewer 4 Report

Thank you for sending me the revised version of this manuscript. The study is now clearer and both the study design and the degree of relevance of the analysis are more transparent compared to the first version.

Comments:

- In addition, I suggest to add to the paper a clarification, which ELEMENTS are specific to OR in comparison to ODLC. This might allow to draw conclusions for contexts outside of Japan. For example, I imagine that the degree of multiprofessional/interdisciplinary collaboration and the focus outside of ADL support might be two elements that are specific to OR. This could provide a lesson for other countries in that a focus on reablement / interdisciplinary therapeutic support might achieve better results than conventional ADL support.

- The conclusions should focus more on these generalizable elements rather than sum up the results (that belong to the results section).

- Also, please explain the abbreviation BPSD in the text. It is not clear what it stands for unless I missed it.

Author Response

Dear Reviewer 4

Thank you for your thoughtful and constructive feedback you regarding our manuscript. We have made revisions based on your comments and suggestions, as shown below. Your original comments are indicated in italics for reference, and our point-by-point responses are given below.

Thank you for sending me the revised version of this manuscript. The study is now clearer and both the study design and the degree of relevance of the analysis are more transparent compared to the first version.

Comments:

- In addition, I suggest to add to the paper a clarification, which ELEMENTS are specific to OR in comparison to ODLC. This might allow to draw conclusions for contexts outside of Japan. For example, I imagine that the degree of multiprofessional/interdisciplinary collaboration and the focus outside of ADL support might be two elements that are specific to OR. This could provide a lesson for other countries in that a focus on reablement / interdisciplinary therapeutic support might achieve better results than conventional ADL support.

Response:

Thank you for your comment on this point. We have added this section to the discussion. Kindly see this addition on p. 13, line 295-299.

- The conclusions should focus more on these generalizable elements rather than sum up the results (that belong to the results section).

Response:

Thank you for pointing this out. We have made these revisions based on your comment (p. 14, line 358-363).

- Also, please explain the abbreviation BPSD in the text. It is not clear what it stands for unless I missed it.

Response:

 Thank you for pointing this out. We have revised to add the explanation to the introduction (p. 2, line 69-70).